# OpenReview forum: "Learning to Align and Act: Cross-Modal Gating and Multimodal Reward Shaping for Web Agents"
_ICLR.cc/2026/Conference — ICLR 2026 Conference Withdrawn Submission_

### Official Review · Reviewer_2kwq · 2025-10-31

**Soundness:** 3
**Presentation:** 2
**Contribution:** 2
**Rating:** 4
**Confidence:** 3

**Summary:**

This paper proposes a reinforcement learning framework for web agents that integrates two main innovations: cross-modal attention gating and multimodal reward shaping. The gating mechanism adaptively balances visual and textual inputs (DOM structures and rendered screenshots) at each timestep, aiming to mitigate the misalignment problem inherent in web environments. Complementing this, the reward shaping approach decomposes sparse terminal rewards into intermediate signals derived from visual and textual feedback, guiding the agent more effectively during training.

Empirical results on MiniWoB++, WebShop, and Mind2Web show moderate but consistent improvements in success rates and sample efficiency. The paper claims these gains arise from the synergy between the gating and shaping components, supported by ablation studies and visualization analyses of attention behavior during task execution.

**Strengths:**

The paper’s clarity and implementation quality are strong. Its methodological formulation is concise and intuitive, with a well-illustrated pipeline and clearly defined objectives. The introduction of gating within the web agent context is an original extension of known attention control mechanisms, addressing a meaningful practical issue—DOM–visual misalignment—common in browser-based automation. The integration of modality-specific reward signals with the fusion process also demonstrates a thoughtful, application-oriented design.

Experimentally, the results indicate measurable benefits in convergence speed and task completion, supported by diverse metrics beyond mere task success rate. The narrative is coherent, the visualizations effectively communicate the model’s behavior, and the overall presentation is polished, which enhances readability and interpretability.

**Weaknesses:**

Despite these merits, the conceptual novelty remains somewhat limited. The gating mechanism itself is not new, and its justification over more expressive fusion techniques—such as cross-attention, mixture-of-experts routing, or adapter-based fusion—is not adequately discussed or empirically validated. The reward shaping method also appears heavily dependent on this gating architecture, which reduces generality and raises questions about whether the method would extend to other multimodal RL contexts.

Furthermore, the baselines chosen for comparison do not convincingly represent the current state of the art in web agents. Modern architectures like WebArena or WebAgent-R1, as well as retrieval-augmented or self-correcting approaches, are cited but not included in the experiments. The ablation studies, while informative, focus on component removal rather than alternative design choices, leaving open whether the proposed components are truly optimal. Details on detector and validator implementation are also sparse, limiting reproducibility.

**Questions:**

The authors should clarify the theoretical and empirical justification for choosing gating over more advanced multimodal fusion mechanisms. Comparisons against cross-attention or MoE-based fusion would significantly strengthen the argument. Likewise, it would be useful to analyze whether the proposed reward shaping scheme is potential-based and thus policy-invariant; otherwise, it might bias the learning process or encourage subgoal overfitting.

Adding results on WebArena and including stronger baselines would help contextualize the reported performance gains. Detailed implementation information about the visual change detector and text validator modules should be provided, as these elements critically affect the shaping signal. Finally, exploring whether the gating mechanism generalizes to tri-modal settings or token-level gating would demonstrate broader applicability and robustness.

---

### Official Review · Reviewer_NCM7 · 2025-11-01

**Soundness:** 2
**Presentation:** 1
**Contribution:** 1
**Rating:** 2
**Confidence:** 4

**Summary:**

This paper aims to solve two core problems in web-based RL agents: cross-modal misalignment and reward sparsity in long-horizon tasks. This paper proposes a framework that combines a cross-modal attention gating mechanism to dynamically weight visual and textual features based on task context , and a multimodal feedback-driven reward shaping scheme that converts sparse terminal rewards into dense, step-level signals by detecting UI changes and validating text . Experiments are conducted on MiniWoB++, WebShop, and Mind2Web, claiming improvements in task success rate and sample efficiency.

**Strengths:**

1. The two problems addressed by this paper, cross-modal fusion  and reward sparsity, are critical problems in building robust web agents today.
2. The proposed method evaluated across three benchmarks of varying scale and complexity (MiniWoB++, WebShop, Mind2Web) and provide ablation studies to analyze the two main components.

**Weaknesses:**

1. The visual reward ($R_{vision}$) is defined as detecting "UI state changes (e.g., pop-ups, button color)". This is a problematic assumption. An agent can easily trigger UI state changes by "randomly clicking" (e.g., clicking ads, opening incorrect dropdown menus). Rewarding state change would encourage this type of non-goal-oriented exploration rather than meaningful task progression. The paper does not clearly state how it distinguishes "beneficial" state changes from "meaningless" ones.
2. The textual reward ($R_{text}$) is similarly ill-defined. It is described as "validates field content or label alignment" or "form field correctness, label matching". The paper provides no detail on how this is implemented. Does this validation require an external oracle? Or does it assume the benchmark environment provides this (which is common in many benchmarks, but this is a strong assumption not discussed)? This ambiguity makes the contribution non-reproducible and difficult to evaluate for its true effectiveness.
3. The framework's core architecture uses standard visual (e.g., ViT) and textual (e.g., BERT) encoders, which is an extremely common setup in multimodal research. The paper's claimed novelty lies in the "cross-modal attention gating". However, this mechanism is a simple gated weighted sum generated by a small controller. This adaptive weighting is a straightforward application of gating units and, as acknowledged in the related work, does not conceptually advance beyond existing adaptive fusion techniques.
4. Poor writing and presentation quality. The quality of the figures is low, and there is significant redundancy.

## Minor Issues
1. Appendix Section A.1.1 ("Evaluation Metrics") and Section A.1.4 ("Evaluation Metrics")  are nearly identical, word-for-word repetitions.
2. The paper states it uses ResNet-50/ViT-B/16 for vision and BERT-base/HTML-aware Transformer for text. However, the main results tables (e.g., Table 1) do not specify which combination of backbones was used to achieve the reported SOTA results, making reproduction and fair comparison difficult.

**Questions:**

See Weaknesses

---

### Official Review · Reviewer_SoGH · 2025-11-03

**Soundness:** 2
**Presentation:** 1
**Contribution:** 2
**Rating:** 2
**Confidence:** 5

**Summary:**

This paper addresses two critical challenges in training reinforcement learning agents for web navigation: cross-modal misalignment and reward sparsity. The authors propose a novel framework that integrates a cross-modal attention gating mechanism with a multimodal reward shaping scheme. The gating controller dynamically adjusts the influence of visual (screenshot) and textual (DOM) representations based on the task context, allowing the agent to prioritize the most relevant modality at each step. Complementing this, the reward shaper converts sparse terminal rewards into dense, step-level feedback by detecting intermediate signals of progress, such as UI state changes from the visual stream and content validation from the textual stream. These two components are designed to work synergistically, where the dense rewards provide a strong learning signal for the gating controller, and the adaptive gating ensures the agent's focus aligns with the source of the reward. The authors validate their approach through extensive experiments on three diverse benchmarks—MiniWoB++, WebShop, and Mind2Web, while demonstrating significant improvements over existing methods. The proposed agent achieves a 6-8% higher task success rate and reduces sample complexity by 44%.

**Strengths:**

[S1] This paper studies a highly popular topic (web agent) and tries to address important research questions (how to leverage both text and image modalities).

[S2] The proposed method (gating + reward shaping) works better among the baselines (RAG/Toolformer/PPO/SAC) in various aspects.

**Weaknesses:**

[W1] There are not sufficient discussion/comparison to the previous literature in this domain; Humphreys et al., (https://arxiv.org/abs/2202.08137) proposed multimodal computer control agents and achieved human level performance with RL in MiniWoB++. Furuta et al., (https://arxiv.org/abs/2305.11854) and Shaw et al., (https://arxiv.org/abs/2306.00245) proposed to leverage multimodal information with the recent multimodal LLM architectures. Zheng et al., (https://arxiv.org/abs/2401.01614) proposed the web agent based on GPT-4V. WebRL (https://arxiv.org/abs/2411.02337) and  UGround (https://arxiv.org/abs/2410.05243) proposed RL training or multimodal architecture to resolve web navigation. Please consider to compare the performance and discuss the difference and relevance with those works.

[W2] "Preliminaries" section is really needed. It is quite unclear which variable/subscripts stands for what. The current paper significantly relies on the reader's knowledge.

[W3] The proposed architecture is quite similar to WebShop (https://arxiv.org/abs/2207.01206); ResNet + BERT. Moreover, there are no discussion or explanation about why the authors did not use LLMs or VLMs as a backbone models, which are already used in recent web agent works (WebRL: https://arxiv.org/abs/2411.02337,  UGround: https://arxiv.org/abs/2410.05243).

[W4] The proposed method is not evaluated in recent and more advanced multimodal web benchmark such as Visual WebArena (https://arxiv.org/abs/2401.13649). I think the results on Visual WebArena are more valuable than including the results in MiniWoB++ for the recent web agent research community.

[W5] There are many unclear points in evaluations and settings. Please see **Questions** section below.

[W6] There are many points to be improved in the formatting and description in the figures. (1) "Conference Submissions" in the title. (2) unclear upperscript $o_{tv}^{1}$ in Figure 1, (3) unnecessary parenthesis in Figure 1, (4) "current stat" in Figure 1, (5) unclear right arrow from "Gross-Modal Gating Controller" to $a_t$, (6) unclear node of $a_t$ with "reward signal" in Figure 1, (7) "sigoiod" in Figure 2, (8) unnecessary parenthesis in Figure 2, (9) unclear right arrow with "RL Optimization Loop" in Figure 3, (10) lack of "R_t" in "Shaped Reward Computation" node in Figure 3, (11) we cannot read some text in Figure 6 by the overlap with graph legends, etc.

**Questions:**

[Q1] In WebShop (https://arxiv.org/abs/2207.01206), the reported task success rate was at most 28.7%. However, this paper reports 60-75% (in Table 1). I think the base architecture is mostly the same as what WebShop paper proposed. Could you give us detailed explanations?

[Q2] Also, in Mind2Web (https://arxiv.org/abs/2306.06070), the reported task success rate was at most 7.1%. There are huge  gaps compared to this paper. Could you also give us detailed explanations?

[Q3] Could you provide the details of "RAG-style Agent" and "Toolformer-style Agent"?

[Q4] How many tasks & what kinds of tasks are you using in MiniWoB++? In prior works (https://arxiv.org/abs/2306.00245, https://arxiv.org/abs/2305.11854, https://arxiv.org/abs/2202.08137, https://arxiv.org/abs/2303.17491), there are variety of subsets, and the choices of tasks can lead to different average success rates.

[Q5] Could you explain more about attention-entropy? Where (+ which layer, what kinds of tokens, how) do you measure the entropy of attention? In the current paper, it is hard to parse that information.

---

### Official Review · Reviewer_7wGf · 2025-11-07

**Soundness:** 2
**Presentation:** 2
**Contribution:** 2
**Rating:** 2
**Confidence:** 2

**Summary:**

This paper proposes a web-RL framework that fixes visual–text misalignment and sparse rewards using dynamic cross-modal gating and multimodal reward shaping. The gating module adaptively balances vision and DOM features, while the reward shaper turns sparse terminal rewards into dense visual/textual feedback. Across MiniWoB++, WebShop, and Mind2Web, the method improves success rates by 6–8% and cuts sample complexity by 44%. Ablations show both components are complementary and crucial for robust web agents

**Strengths:**

- Giving the agent the ability to shift between vision and web (DOM) information is an interesting innovation

- Densifying the reward facilitates the learning problem

**Weaknesses:**

- The reward shaping procedure depends on manually defined visual/state-change detectors and text validators. This limits generalization to pages with unusual layouts, noisy DOMs, or non-standard UI behaviors.
- Limited/no evaluation in larger "in-the-wild" large scale websites with complicated UIs and more sophisticated layouts / tasks.

**Questions:**

Have the authors evaluated their method on more complicated website datasets?

The paper formatting is a bit strange, with much whitespace left above and below tables. Could the authors improve formatting?

---

### Note · Authors · 2025-11-19

I have read and agree with the venue's withdrawal policy on behalf of myself and my co-authors.